# New Arc Stability Index for Industrial AC Three-Phase Electric Arc Furnaces Based on Acoustic Signals

**DOI:** 10.3390/s20236840

**Published:** 2020-11-30

**Authors:** Juan Guerra-Serrano, Angel Sánchez-Roca, Guillermo González-Yero, Mario C. Sánchez-Orozco, Mercedes Pérez de la Parte, Emilio Jiménez Macías, Julio Blanco-Fernández

**Affiliations:** 1Integral Automation Company (CEDAI), L.Ortiz No.36 Esq.Lora, Las Tunas 75100, Cuba; juangs87@nauta.cu; 2Faculty of Mechanical Engineering, Universidad de Oriente, Ave. Las Américas s/n, Santiago de Cuba 90900, Cuba; sanchez@uo.edu.cu (A.S.-R.); mario@uo.edu.cu (M.C.S.-O.); 3Development and Innovation Group, ACINOX Las Tunas, Zona Industrial, Circunvalante Norte km 3½, Las Tunas 75100, Cuba; guillermo@acinoxtunas.co.cu; 4Mechanical Engineering Department, Universidad de La Rioja, Ave. de la Paz, 93, 26006 Logroño, La Rioja, Spain; mercedes.perez@unirioja.es; 5Electrical Engineering Department, Universidad de La Rioja, Ave. de la Paz, 93, 26006 Logroño, La Rioja, Spain; emilio.jimenez@unirioja.es

**Keywords:** stability index, electric arc furnace, arc stability, acoustic emission, signal processing

## Abstract

This research proposes a new index to evaluate the stability of the melting process, in three-phase electric arc furnaces (EAFs), based on the acoustic signals generated during the different stages of the casting. The proposed stability index is obtained by characterizing the time and frequency domain of the acoustic signals. During EAF monitoring, acoustic signals were acquired using a microphone coupled to an NI USB-9234 acquisition system. To validate the results, the voltage and current signals were measured with the aid of a Circutor AR6 power analyzer for three-phase electrical networks. The results showed that the acoustic signal energy in the frequency range of 1 to 12 kHz can be used as an indicator of the process stability in the EAF. Finally, the validity of the proposed stability index is evaluated from the process characterization using the harmonic distortion analysis methods and the dynamic U-I characteristics of the arc voltage and current signals. The results obtained demonstrated the effectiveness of the proposal and constitute a starting point for advances in the automatic control of the process in the EAF, from the acoustic signals.

## 1. Introduction

In the metallurgical industry, the use of three-phase alternating current (AC) electric arc furnaces (EAFs) plays an important role in international steel production. In the EAF, the melting of the material is done from the high temperature generated mainly by the electric arcs. These EAFs are large consumers of electrical energy and therefore, studies focused on the control of its variables to reduce consumption and increase efficiency are very diverse [1,2,3]. In this sense, it is considered that the results achieved to measure the stability of the arc in the EAFs are very important, and therefore constitute an open research problem.

Currently, the main parameters used for arc control in three-phase EAFs are extracted mainly from the arc voltage and current signals in each of the three phases. The main objective is the regulation of the arc height maintaining the control of the voltage-current ratio for a desired operating point in the transformer. [4]

The characterization of voltage and current harmonics is widely used for the control of EAFs. In this sense, Uz-Logoglu et al. [5] proposed a method to detect the variations in the time of harmonics and inter-harmonics, with the objective of controlling the fluctuations in nonlinear industrial loads. Jopri et al. [6] used the time-frequency diagrams (spectrograms) to obtain classifiers that allow evaluating the contribution of harmonics and inter-harmonics of voltage and current signals in the EAF. Dehkordi et al. [7] proposed a new method to evaluate the quality of the foamy slag in the EAF. In this case, they used a neuro-diffuse inference system and fuzzy logic from the analysis of current harmonics. Specifically, they demonstrated that the total distortion of the seventh current harmonic and the current unbalance of the three phases guaranteed the correct evaluation of the system.

During the operation of the EAF, electric arcs generate changes in sound pressure causing a high level of noise. EAFs are a source of noise pollution with a negative impact on the health of the personnel operating the furnace [8,9]. Usually, during furnace operation, the experienced operators determine the operation of the EAF, the stages of the melting process, and the quality of the foamy slag from the sound generated by the EAF. Although, in practice, measurements of voltage and current signals are mostly used for monitoring, control, and modeling of the EAF [10], previous studies [8,9,11,12] have shown that the acoustic emission (AE) signal can be used to characterize the operation of the EAF.

The acoustic signal generated during the casting process is mainly affected by the irregularities in the stability of the arc. Attempts to use the acoustic signals to characterize the process in the EAF are not recent. Since the previous century, Drouet and Nadeau [13] established in their studies the relationship between the time integral of the acoustic signal generated by the process and the power of the arc column. On the other hand, Sinha and Gupta [14] monitored a monophasic EAF, at the laboratory scale, to evaluate the influence of the increase of the arc current on the acoustic emission levels in the furnace. In this case, they showed that the sound levels decreased with time as the load in the furnace melts and increased with the increase of the arc height.

Modeling the process from experimental data for arch height classification tasks based on Kernel Fisher discriminant analysis was proposed by Burchell et al. [15]. In this case, the authors performed studies in a 60 kW DC EAF and extracted AE signal characteristics under stable conditions, to evaluate 5, 15, and 25 mm arc heights. Chen et al. [16] made a numerical simulation of the acoustic waves generated by alternating current arcs. In this case, the relationship between the derivative of the arc power (dP/dt) and the sound pressure was demonstrated. The high relationship between the arc power derivative waveforms and the sound pressure signal was also demonstrated. Cherednichenko et al. [9] made a simulation of the process in the EAF from a mathematical model with the aim of describing the acoustic characteristics of sound in the EAF and its energy. The authors of this paper proposed to keep the highest amplitudes of the AE signals in the frequency range of 5–150 Hz.

One of the stages of the melting process in the EAF is the phenomenon of foamy slag. Foamy slag is formed from gas bubbles developed due to the injection of gas into the molten bath. The formation of the foamy slag contributes to the improvement of the efficiency of the EAF during the final stage of heating and refining. Regularly, the quality of the foamy slag is determined by the experience of the operator from the sound generated during this stage of the process. Due to its importance, the main studies that relate the process in the EAF with the vibro-acoustic signals are focused on the analysis of this final stage of the casting.

The effect of the foamy slag process on the sound generated by the EAF has also been evaluated by Matschullat et al. [17]. The authors, using three accelerometers, monitored, the vibrations on the walls of an EAF in order to control the foam slag process from the setting of the carbon injection using a fuzzy controller. As stated by Komarov et al. [18], the intensity and frequency of acoustic waves below 1.3 kHz, generated during this stage of the process, can be used as an indicator of the rate of variation of the foamy slag. Raichel et al. [19] showed that the covering of the arc by the foamy slag reduces considerably the sound levels and variations in the range of 95 to 90 dB.

The most recent research on the application of acoustic signals in an EAF was focused on evaluating the feasibility of their use to control the different stages. Cherednichenko et al. [8] in a laboratory furnace placed three sensors to measure the vibrations which allowed the detection of the EAF’s different operating conditions. The feasibility of using AE signals to detect the different stages of the process in the EAF was also described by Fu et al. [12]. In this case, the method used was the LPC (linear predictive coding) and a principal component analysis (PCA) algorithm to identify the different stages of the process. 

Recently, Fu et al. [11] proposed a control system based on digital images and acoustic signal processing, which focused on splash suppression in the EAF. The authors concluded that unstable arcs generated more harmonics in the acoustic signal. They also showed that the second harmonic of the signal had a higher amplitude than the first harmonic when the system operated under unstable conditions.

A proposal for an arc stability index in the EAF was presented by Vicente et al. [2]. In this case, the authors proposed a non-invasive method to obtain an index based on the measurements of the magnetic field in a direct current (DC) EAF, using an array of Hall effect sensors in three coordinated axes. The validity of the index was verified from its comparison with the acoustic signal generated by the EAF in the frequency range up to 1 kHz. The proposed stability index showed a strong correlation with the acoustic signals.

From the previous analysis, it can be deduced that there are numerous articles that discuss the use of the acoustic signal in the EAF, both theoretically and experimentally. Most of the publications focus on the modeling of the acoustic signals from the analysis of their correspondence with the voltage and current signals of the arc. Generally, the presence of interferences from other EAFs and industrial noise in the vicinity, which can affect the correct identification of the process, is considered as the main limitation for the application of the acoustic signal. Taking into account the previous studies retrieved from the consulted literature, this article analyses and offers a detailed explanation of the methodology and results achieved, from the processing of acoustic signals, to obtain a new stability index of the arc. This proposal is an important step towards a new approach for future applications in automatic process monitoring and control in the EAF.

## 2. Materials and Methods 

### 2.1. System Description

The experiments were performed in a Danieli^®^ [20] three-electrode EAF, powered by a three-phase alternating current (AC). The main characteristics of the EAF used to make the tests are shown in Table 1.

Figure 1 shows a diagram of the experimental facility used to conduct the tests. Taking into account the environment around the EAF, the equipment to perform the measurements was positioned to collect the data from two control cabins of the furnace.

### 2.2. Acoustic Signal Acquisition and Processing

The acoustic signals were acquired using a G.R.A.S^®^ [21] type 46AG "free field" microphone with a sensitivity of 12 mV/Pa, frequency range: 3.15 Hz to 40 kHz, and a dynamic range of 17 to 146 dB. The microphone signal was coupled to a G.R.A.S^®^ amplifier, model 12AQ, and digitized using a NI-USB 9234 data acquisition card with a sampling frequency of 51.2 kS/s. The measurement chain was calibrated with the aid of a G.R.A.S^®^ 42AP intelligent pistonphone. The acquired signals were stored using a program specifically designed for this purpose in LabView^®^ [22] and processed in Matlab^®^ [23]. Initially, the voltage levels delivered by the signal conditioner were converted to sound pressure (Pa) considering the sensitivity of the microphone.

#### 2.2.1. Industrial Noise Characterization

One of the main limitations of the use of acoustic signals for the characterization of processes on an industrial scale is environmental noise. Noise in an industrial environment is complex and can affect measurements, reducing the accuracy of the proposal [7]. EAFs generally work in conjunction with ladle furnaces and other process components in the steel plant, whose sound can mask the results of acoustic measurements. Considering this limitation, a noise characterization was performed in the EAF environment. In order to perform the industrial noise characterization, the acoustic signal was monitored in three operating conditions: both furnaces (EAF and ladle furnace) turned off to characterize the environmental noise, the ladle furnace in operation and the EAF turned off and, finally, the condition with both furnaces in operation.

The industrial noise analysis was made by signal processing in the frequency domain. To study the frequency bands associated with industrial noise, the time-frequency diagrams (spectrograms) of the acoustic signals in the range of 0.2 Hz to 25 kHz were used. These analyses allowed us to determine which frequency bands of interest provided useful information and which were associated with industrial noise and did not contribute to the study [24]. The spectrograms obtained from the application of the Fast Fourier Transform (FFT), were a representation of the energy of the acoustic signal with respect to time and frequency. This method reflected the variations of harmonics over time. To obtain the spectrogram, the signal was divided into segments of equal length, using in this case a Hanning window.

Equation (1) shows the mathematical expression for obtaining the spectrogram of the signal.
(1)$$F\left(\tau ,\omega \right)=wFFT\left\{x\left(t\right),w\right\}\left(\tau ,\omega \right)=\;\overset{\infty }{\underset{n=-\infty }{\sum }}x\left(t\right)\;.w\left\{t-\tau \right\}e^{-i\omega t},$$
where: *x*(*t*) is the acquired acoustic signal, *w* is the Hanning window, *τ* and *ω* represent the time and frequency respectively of the Fourier transform of the analyzed window.

The analyses of the spectrograms for the three industrial noise conditions allowed us to define the frequency band with useful information to develop the new process stability index. Finally, for the implementation of the stability index, a bandpass filtering was performed in the frequency range of interest, in order to guarantee immunity to the ambient noise.

#### 2.2.2. Characterization of the Acoustic Signal in the Time Domain

In addition to the spectral components, it was necessary to conduct the analyses in the time domain to detect regularities or irregularities in the process, as an indicator of stability in the frequency band of interest. The filtered acoustic signals were processed in the time domain to extract characteristics that supported the establishment of the new process stability index. For the characterization in the time domain, the signal descriptors were calculated: root mean square (RMS), the sound level in dB, peak-to-RMS ratio of the signal, and signal energy (E). The total energy of the acoustic signal was calculated by applying Parseval’s theorem and integrating, with respect to time, the square of the signal from a cumulative trapezoidal numerical integration, as shown in Equation (2).
(2)$$E=\overset{t}{\underset{0}{\int }}{\left|x\left(t\right)\right|}^{2}dt,\;\;$$
where: *x*(*t*) is the acoustic signal in the time domain.

The peak-to-RMS ratio was determined from the relationship between the highest absolute peak value of the acquired sound signal and the RMS value of the *x*(*t*) signal according to Equation (3).
(3)$$peak-RMS\;ratio=\frac{{\left\|x\right\|}_{\infty }}{\sqrt{\frac{1}{N}\sum_{n=1}^{N}{\left|x_{n}\right|}^{2}}}\;,\;\;\;$$
where: n is the number of samples of the acoustic signal; ${\left\|x\right\|}_{\infty }$ the infinite standard of the *x*-vector, which is defined as: ${\left\|x\right\|}_{\infty }=max\left(\left|x_{i}\right|\right)$ i = 1…, *n*.

An analysis of variance (ANOVA) was performed to determine the significance levels of the acoustic signal descriptors in the time domain to select the appropriate one for the characterization of the stability of the casting process in the EAF.

### 2.3. Industrial Validation of the Proposal

To validate the new proposed process stability index in the EAF, the results obtained were compared with three different established methods of process stability analysis used by other authors [5,25,26,27] from the analysis of voltage and current signals.

#### Acquisition and Processing of Arc Voltage and Current Signals

The voltage and current signals of each phase in the EAF were monitored during all stages of the casting. They were acquired by connecting a three-phase Circutor AR6 [28] power analyzer for electrical networks to the EAF power lines, as shown in Figure 1. The signal sampling frequency was 1 S/s. The signals were stored during all casts for further processing.

One of the methods used to evaluate stability was from the diagrams of dynamic characteristics of the arc voltage (U) versus arc current (I) in each of the phases. The method used was based on the determination of the coefficient of variation (CV) of the areas between each of the phases of the electrodes. A uniform and stable process generated very similar U-I loops with low CV. Otherwise, high CVs were associated with instabilities in the process.

The frequency-domain processing of I and U signals from the Fast Fourier Transform was used for the detection of signal harmonics. Other methods, such as the total harmonic distortion (THD) of the current between phases and the amplitude of the second current harmonic, were also evaluated to estimate the validity of the new proposed stability index.

## 3. Results and Discussion

For each of the casts monitored in the EAF, the melting process took approximately 50 min and was performed as follows: in the first stage, the first basket of scrap was loaded into the furnace and power was applied to remove the volume not occupied by the scrap to allow the addition of another basket (in this case the process was repeated until the addition of a fourth basket that guaranteed the total mass according to the capacity of the EAF). Due to the state of the scrap, the height of the arc at this stage had undergone great variations that caused an increase and irregularities in the levels of the acoustic signal generated (Figure 2a). 

During the second stage, the arc showed a more stable behavior due to a greater amount of molten metal in the metal bath, so the sound emission levels decreased as a result of less variation in the height of the arc (Figure 2b). Once the total melting of the last basket had been reached, it was possible to move on to the third stage of heating and refining of the steel. In this stage, the levels of the acoustic signal decreased significantly (Figure 2c), due to the flat nature of the metal bath and the performance of the foaming slag process, which increased the efficiency of the heating process. This stage ended when the desired carbon level was reached and the bath reached a temperature of 1600 °C. 

To conduct the studies, three operating conditions of the furnace were selected: one unstable, which corresponded to the first phase (unmelted); one semi-stable, which corresponded to the second phase (semi-molten), and; one that was stable, which related to the third stage of the process (molten). Figure 2 shows the acoustic signals acquired for the three conditions studied. 

As shown in Figure 2, the overall acquired acoustic signals showed irregularities and high levels (Figure 2a,b) associated with instabilities related to shutdowns and restarts of the arcs as a result of the irregularities of the scrap metal. The signal shown in Figure 2c has a more regular behavior and low levels of amplitude by the presence of more stable arcs and a lower height in a metal bath which was virtually flat. Table 2 shows the statistical descriptors extracted from the signals analyzed.

The comparative analysis of the data presented in Table 2 allowed us to conclude that the descriptors that experienced the greatest variation during the three stages analyzed were RMS, sound levels, and acoustic signal energy. Figure 3 shows the mean graphs that demonstrate the effect of changes in the experimental conditions analyzed on the behavior of these signal descriptors.

The energy of the acoustic signal was the descriptor that showed the greatest variation in response to changes in process conditions. Despite the behavior shown in Figure 3, these acoustic signals were affected by sound pressure levels associated with the operation of the ladle furnace and other manufacturing processes in the EAF environment. This noise in the EAF environment was characterized to eliminate its influence on the final results, thus eliminating one of the main limitations of the methods that performed characterization using acoustic signal processing techniques.

### 3.1. Industrial Noise Characterization

Figure 4a shows the acoustic signal in the time domain of industrial noise for the first condition evaluated, which corresponded to the noise around the EAF with the ladle furnace and EAF turned off. A low level of ambient noise was observed, which may be associated with continuous casting processes, cranes, engines, scrap transport, among others. The frequency distribution shown in Figure 4b demonstrates the presence of spectral components of noise in the band from 10 Hz to 1 kHz. The higher level harmonics were mainly associated with electromagnetic noise induced by the power supply frequency (60 Hz). The temporal evolution of the spectral bands shown in the spectrogram of Figure 4c allowed us to corroborate the absence of harmonics in the high-frequency bands with the highest amplitudes located at frequencies below 1 kHz.

Figure 5 shows the graphs that allowed the effect of industrial noise to be evaluated when working under an operational condition where the ladle furnace was functioning and the EAF remained off. Under these conditions, the levels of the acoustic signals experienced an increase (Figure 5a), caused by the proximity of the EAF to the ladle furnace. As the process in the ladle furnace was more stable, the amplitude levels of the harmonics in the frequency domain showed a unique behavior (Figure 5b), with amplitude levels 10 times higher than those detected for the previous operational condition (both furnaces shut down).

As shown in Figure 5b, the operation of the ladle furnace caused an increase in the levels of the spectral bands. In this case, as shown by Fu et al. [11], the frequency of the fundamental harmonic of the acoustic signal was twice the frequency of the EAF feed. The amplitude of the main harmonic of the 120 Hz acoustic signal was lower than the second harmonic of 240 Hz. This behavior was typical of a stable process, characteristic of the ladle furnace. Any unbalance between the arcs caused an increase in the amplitude of the first harmonic (120 Hz) of the acoustic signal. [13]

Figure 5b also shows harmonics with frequencies below 50 Hz, which, according to studies by Cherednichenko et al. [8], are associated with electromagnetic vibrations of the electrodes, electrode clamps, among others. The frequency spectrum of the acoustic signal corresponded only to the first 18 seconds of the acquired signal. The last seconds were not taken into account, since in this time interval, it was decided to connect the EAF’s power supply to evaluate its effect on the frequency domain.

Figure 5c shows the spectrogram of the industrial noise signal. In this case, during the first 18 s, only significant spectral components were observed at frequencies below 1 kHz. In the 1–12 kHz frequency band, spectral components were only observed after 18 s of measurement. This time-lapse corresponded to the beginning of the casting in the EAF operating under conditions of low arc stability (unmelted).

### 3.2. Proposal for a New Arc Stability Index

The results of the noise characterization were considered for the definition of the proposed new Acoustic Signal Based Arc Stability Index (*AESI_EAF_*). The harmonics below 1 kHz associated with the operation of the EAF could be affected by noise from the industrial environment and decrease the reliability and accuracy of the method.

Three conditions related to the stages defined above were selected for the establishment of the stability index. Figure 6a shows the acoustic signals generated in the EAF for an unstable (unmelted), semi-stable (semi-molten), and stable (molten) condition. During the initial phase, the interaction between the arcs generated instability and therefore an alteration in the pressure levels within the EAF. As stated by Cherednichenko et al. [9], these instabilities generated acoustic waveforms with great variability and maximum sound pressure levels caused by the discharges of the arc column and the circulation of the electric current through the fragments of the furnace charge.

Figure 6b shows the spectrograms of the signals in the different stages of the process. In this case, spectral components were observed in the band from 1‒12 kHz, which decreased as the scrap was completely molten. The arc restart instances generated high levels of the spectral components of the acoustic signal in this frequency band, as a result of the variations in the arc current levels during the arc discharge. This destabilization of the current levels, which generated changes in sound pressure levels, was related to the numerous electrical discharges between the fragments of the charge during the initial stage of the casting [8]. As the charge changed to a liquid state, the metal bath became flatter and the arc tended to present a more stable behavior [29], reducing the distortion levels of the acoustic signals over time. This improvement in arc stability was evidenced by a significant decrease in spectral components in the 1–12 kHz frequency band.

Taking into account the previous analyses, it seemed safe to hypothesize that when the casting process in the EAF operated in an unstable way, the spectral components in the frequency band from 1–12 kHz showed high levels of amplitude and dispersion. On the contrary, greater stability in the process would be characterized by a significant reduction in the levels of distortion and amplitude of the acoustic signal generated in the EAF. To obtain the new stability index, the acoustic signals were filtered using a band-pass filter (BP) with lower cut-off frequency *f_ci_* = 1 kHz and upper cut-off frequency *f_cs_* = 12 kHz, to capture only the acoustic signals free from the interference associated with the previously characterized industrial noise. The advantage of acoustic signal filtering was to eliminate the effect of ambient noise on measurements. To ensure the validity of the proposal, the sampling frequency of the data acquisition system had to be equal to or higher than 25 kS/s. This minimum value guaranteed analysis in the frequency range of interest.

Figure 7 shows the filtering sequence of the acoustic signals to obtain the process stability index. The signals generated during the process in the EAF for the three conditions analyzed (Figure 7a) were filtered (Figure 7b), and a reduction in the levels of signal amplitudes was observed by eliminating the frequency components associated with the power delivered to the EAF and industrial noise.

The behavior of the filtered signals shown in Figure 7b, evidenced the presence of irregularities in the acoustic signal with a presence of higher amplitude peaks for the most unstable conditions (unmelted) that may be associated with short-circuits and restarts of the arcs. In this first stage, the signals showed higher energy in the evaluated frequency range, caused by the erratic movement of the arcs due to the state of the load [15]. The condition of greater stability (molten) presented a filtered signal of less energy and with less dispersion of the peaks associated with the instabilities of the electric arcs. This behavior was associated with the presence of a total fusion of the charge, with arcs of similar height and energy.

Figure 7c shows the behavior of filtered signals in the frequency domain. In this case, the condition of greater instability (unmelted) showed higher frequency peaks. The higher signal energy was related to the solid-state of the load. As the load melted (semi-melted), the amplitudes of the peaks in the spectrum decreased. When the basket was fully fused, and consequently the arcs exhibited more regular behavior, the 1–12 kHz spectral components reduced their levels considerably as a result of the increased process stability in the EAF.

Figure 8 shows the ANOVA graphs that allow corroboration of the above. In all cases, the energy and amplitude levels of the filtered signals in the frequency range of 1–12 kHz decreased as the load in the furnace changed from solid to liquid, making the process more stable.

The behavior described above confirmed that the filtered acoustic signal energy in the range of 1–12 kHz was an appropriate parameter to represent the behavior of the casting process in the EAF. Based on this assessment, it was possible to define the new arc stability index based on the acoustic signal *AESI_EAF_* as: the sum of the filtered acoustic signal energy in the frequency band from *f_ci_* to *f_cs_*, generated in the EAF during the casting. The index is expressed in Pa^2^·s, and is described by Equation (4).
(4)$$AESI_{EAF}=\sqrt{\overset{fc_{s}}{\underset{f=fc_{i}}{\sum }}x{\left(t,f\right)}^{2}}\;,\;$$
where: *x*(*t,f*) is the magnitude of the signal in time in the frequency range defined by *f.*

To evaluate the behavior of the proposed stability index, the filtered signals of the three operational conditions of the process were selected (Figure 9a). In this case, the spectrogram of the signals (Figure 9b) showed high-energy frequency components during the first stage of the process, which decreased as a result of higher arc stability in the EAF. The vertical red lines shown in the spectrogram corresponded to arc restart instances and erratic arcs which indicated instability in the process. The conditions of greater stability did not present these erratic arc conditions and in this case, the frequency components in the 1–12 kHz band were of lower energy.

Figure 9c shows the quantitative values of the *AESI_EAF_* for the three operational conditions studied. During the first stage (unmelted), in correspondence with the high instability of the arcs, the stability index showed high levels and a great dispersion. As time went by, in the next stage (semi-molten), the index levels and their dispersion decreased due to a higher percentage of liquid charge in the furnace. Finally, a decrease in the index values was observed when the process operated with a molten metal bath and very stable arcs whose behavior was evident in a low dispersion of the stability index values in the third stage (molten). Table 3 shows the average values of the stability index and its standard deviation for each of the operating conditions evaluated.

The proposed index allowed the establishment of the overall stability of the casting process in the EAF. However, this stability index was the result of the effect of each of the arcs in each phase. If we took into account that each of the arcs could operate asymmetrically, then, it seemed safe to assume that in the case of three-phase EAFs, it was possible to obtain the effect of each of the three arcs on the global stability index *AESI_EAF_*. For this, it was necessary to modify the index considering the electrical signals of voltage (U) and current of the arc (I). In this case, the stability index *AESI_EAF_* was decomposed into three components, each associated with the effect that this phase induced in the overall stability of the process in the EAF, resulting in Equation (5).
(5)$$AESI_{EAF}=\frac{AESI_{EAF}.\;Z_{1}}{Z_{1}+Z_{2}+Z_{3}}+\frac{AESI_{EAF}.\;Z_{2}}{Z_{1}+Z_{2}+Z_{3}}+\frac{AESI_{EAF}.\;Z_{3}}{Z_{1}+Z_{2}+Z_{3}}\;,$$
where: *Z*1, *Z*2, and *Z*3 are the arc impedances in each phase.

Figure 10 shows the effect of each electric arc on the *AESI_EAF_* stability index levels. The decomposition of the stability index into three components, each associated with a phase, allowed the detection of the operating state of each of the three arcs, therefore, valuable information was obtained to achieve better progress in process control in the EAF. As shown in Figure 10, the individual stability index provided information on the level of stability of each arc. High levels of the index corresponded to unstable conditions, and lower values were associated with greater regularity in the arc. The *AESI_EAF_* index was the sum of each of the components of the three phases. 

### 3.3. Industrial Validation of the Proposed Stability Index

Once the new *AESI_EAF_* index was obtained, it was necessary to verify its validity. In this section, the new stability index proposed was compared with three classical methods of stability analysis in processes of this type. 

#### 3.3.1. Correlation of Stability Index with Harmonic Analysis of Current Signals

The generation of harmonics in the voltage and current signals of the arc was a result of the behavior of the process during its different stages. The distortion of the harmonics was high when the process operated in an unstable way and its levels decreased as it became more stable [4]. This feature allowed the total harmonic distortion (THD) to be used to monitor and control the process. Figure 11a shows a comparison between the proposed stability index *AESI_EAF_* (red) and THD of the current in each of the three phases. The first two stages of the process (unmelted and semi-molten) had the highest THD values and high levels of dispersion between them. In contrast, these values decreased their levels during the final molten basket stage (molten) as a result of greater arc stability associated with the total melting of the furnace charge.

The analysis of the second harmonic of the current signals of the three phases shown in Figure 11b, also had a behavior similar to the *AESI_EAF_* index. It was possible to conclude that the proposed stability index was appropriate for the detection of arc instabilities in the EAF.

#### 3.3.2. Correlation of Stability Index with U&I Signals

For a more complete analysis in this section, a quantitative evaluation of the stability of the process in the EAF from the processing of the U and I signals was made using methods established by other authors [30,31]. These results were compared with the new proposed stability index.

As shown by Liu et al. [10], voltage-current characteristic curves were the most effective way to characterize the behavior of the electric arc in EAF. The analysis of the deviations of the areas of the loops resulting from the U-I curves was a valid method that allowed us to estimate the stability of the arc from the voltage and current signals of the arc, as demonstrated by several researchers. [8,32,33]. Figure 12 shows the behavior of the arc voltage and current signals for each of the phases.

As shown in Figure 12a,b, the cycle shapes of the arc voltage and current signals were not identical. In the initial stage of the process (unmelted), the large instabilities in the arc caused significant variations in the current and voltage waveforms that acquired a non-sinusoidal character due to the discharges in the arcs. This behavior caused the appearance of harmonics with great distortion between them, as shown in Figure 11. The characteristic curves U-I shown in Figure 12c confirmed the existence of loops with very irregular shapes and different areas. In the second stage (semi-molten), lower variability in the voltage and current waveforms was appreciated (Figure 12a,b) and loops of the U-I characteristics (Figure 12c) with a more elliptical shape existed, although with some distortion between phases. For the condition of greater stability (molten), sinusoidal signals were observed with very little presence of harmonics that generated U-I loops with very similar areas. A quantitative analysis of the areas of the loops and the coefficient of variation between them for various periods of the U and I signals is shown in Table 4.

The results shown in Table 4 confirm the above. The coefficients of variation of the areas of the loops showed a decrease as the load was melted in the EAF. Figure 13a shows the behavior of the CV of the areas for the cycles analyzed. There were well-defined groups associated with each of the three operational conditions of the process studied.

Figure 13b shows the relationship between the CV of the U-I loop areas and the values of the *AESI_EAF_* index associated with each condition evaluated. The results obtained demonstrated the validity of the proposal, with a high correlation between the classical method analyzed, based on the U-I signals, and the *AESI_EAF_* arc stability index obtained from the measurements of the acoustic signals generated during the casting process at the EAF.

Finally, Figure 14 shows the real-time behavior of the new stability index *AESI_EAF_*, evaluated for a casting basket. Figure 14a shows the behavior of the filtered signal. In this case, an acoustic signal with the highest levels of amplitude and the presence of transient peaks was observed at the beginning, which occurred due to high energy discharges during shutdowns and restarts of the arc. As time increased, the levels of the acoustic signal decreased until its level at the end of the basket decreased significantly.

Figure 14b shows the behavior of the stability index during the whole process in the basket. In this case, the highest values and dispersions of the index were observed at the beginning of the casting, which decreased as time went by. The graph showed moments where there were high peaks in the *AESI_EAF_* index that corresponded to the moments of the restart of the arc shown in Figure 14a. This behavior showed the sensitivity of the new proposal to irregularities in the stability of the process. The graphs in Figure 14c represent the current THD values obtained from the analysis of the arc currents of the three phases. In this case, a trend similar to the one shown by the proposed index can be seen.

The main contribution of the proposed methodology was from the extraction of information from the acoustic signal in the frequency range from 1 to 12 kHz. The signal energy in this frequency range was unaffected by industrial noise and could be used as an indicator of process stability and as a signal for closed-loop control of the electrode position in the HAE. In the proposal presented, once the acoustic signal was filtered, an analysis was performed in the time domain, which reduced processing times, as it did not require fast Fourier Transform (FFT) calculations. Additionally, it had a lower implementation cost and was more viable than the option of using a THD analysis. Despite the low implementation cost of the proposal, the equipment for the measurement of the acoustic signals and the preparation by the operators could constitute the main limitation for the implementation of the proposal.

## 4. Conclusions

Based on the experimental results obtained from the real-time monitoring of the industrial EAF, it can be concluded that:The proposed new stability index based on the energy of the filtered acoustic signals in the frequency range of 1–12 kHz is an indicator of the process stability and is a simple and valid method for real-time monitoring of the stability of the arcs in the EAF.The filtered signal in the range of 1–12 kHz is immune to the disturbances associated with industrial noise, eliminating one of the main disadvantages of non-invasive methods based on acoustic signals.The decomposition of the *AESI_EAF_* index into three components associated with each phase allows us to determine the incidence of each arc in the global stability of the process, which could lead to better advances in the diagnosis and control of the process in the EAF.Future work will focus on the application of the proposed new index as an electrical signal to improve monitoring, evaluation, decision-making support, and closed-loop control of the process. This would allow an adjustment of arc operating points to ensure greater stability and efficiency in the operation of the EAF.

## Figures and Tables

**Figure 1 sensors-20-06840-f001:**
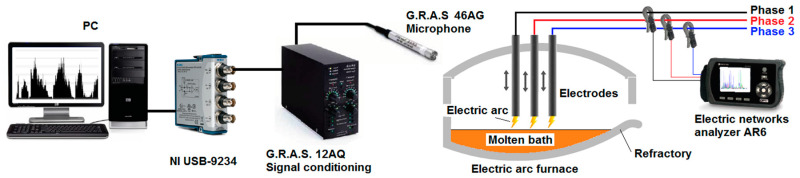
Experimental setup.

**Figure 2 sensors-20-06840-f002:**
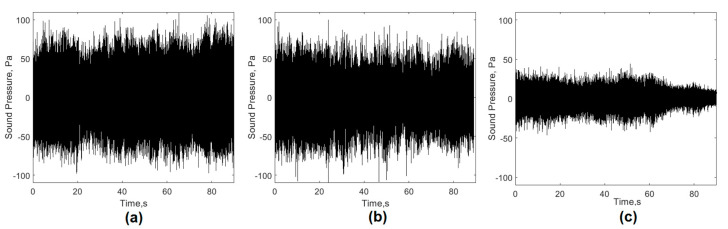
Acoustic signal generated by the EAF during a casting, (TAP = 15). (**a**) unmelted; (**b**) semi-molten; (**c**) molten.

**Figure 3 sensors-20-06840-f003:**
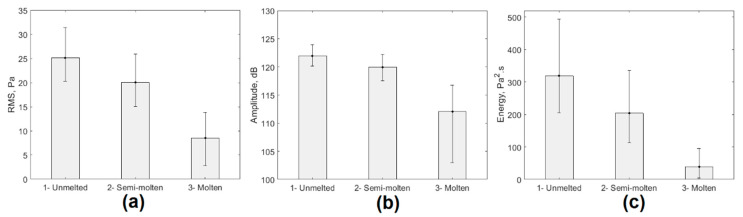
ANOVA of the behavior of the acoustic signal generated by the EAF. (**a**) RMS; (**b**) sound level; (**c**) signal energy.

**Figure 4 sensors-20-06840-f004:**
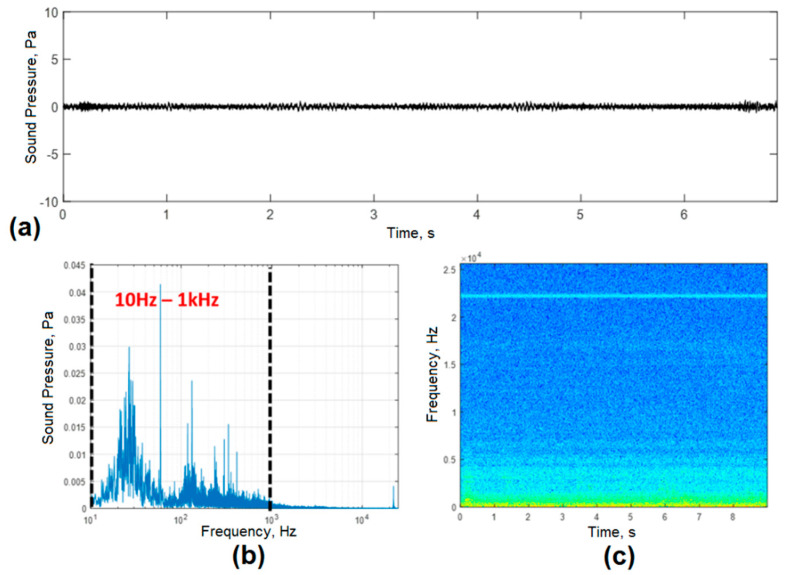
Characterization of industrial noise with the EAF and the ladle furnace turned off. (**a**) acoustic signal; (**b**) frequency distribution of the acoustic signal; (**c**) time-frequency diagram of the evolution of the acoustic signal.

**Figure 5 sensors-20-06840-f005:**
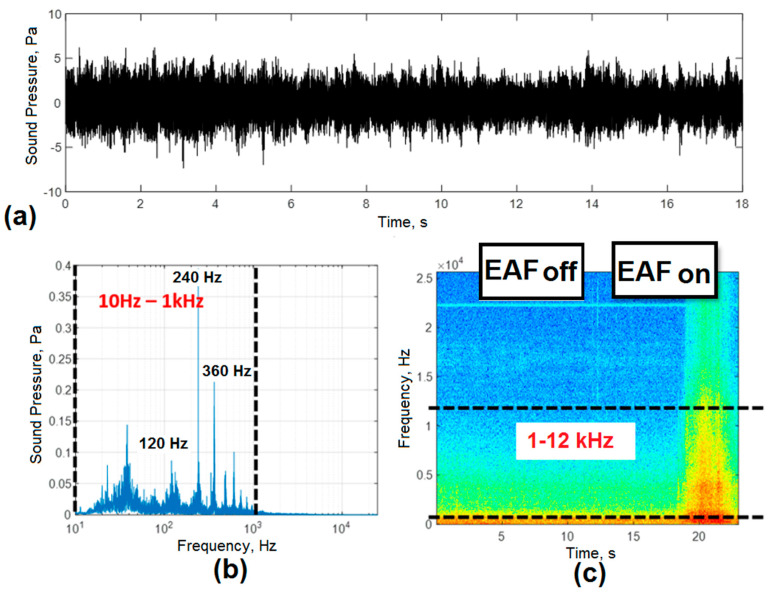
Characterization of industrial noise with the ladle furnace on and the EAF off. (**a**) acoustic signal; (**b**) frequency distribution of the acoustic signal; (**c**) time-frequency diagram of the evolution of the acoustic signal.

**Figure 6 sensors-20-06840-f006:**
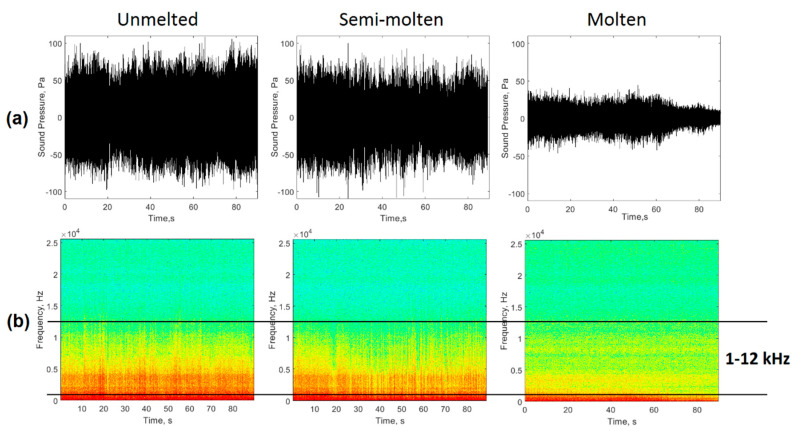
Acoustic signals generated in the EAF during casting. (**a**) acoustic signal; (**b**) time-frequency diagram of the evolution of the acoustic signal.

**Figure 7 sensors-20-06840-f007:**
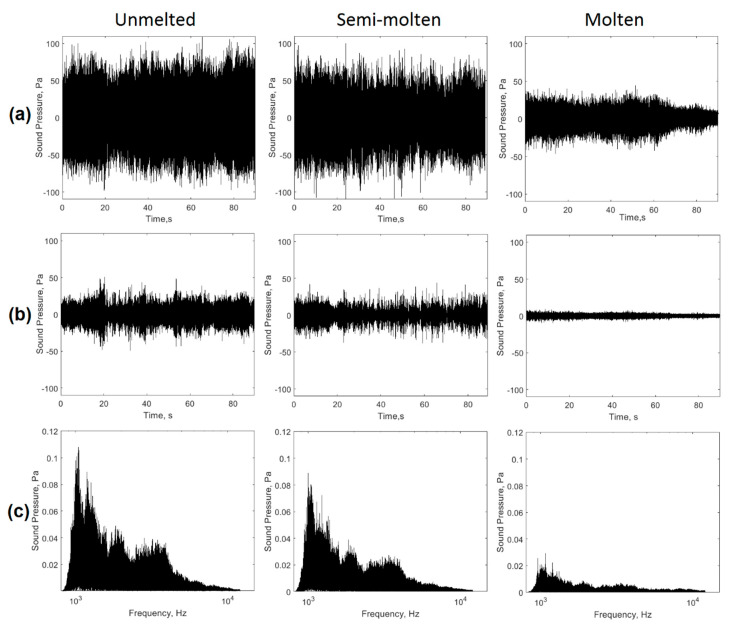
Processing of acoustic signals. (**a**) acquired acoustic signals; (**b**) filtered acoustic signals; (**c**) amplitude spectrum in the range of 1–12 kHz.

**Figure 8 sensors-20-06840-f008:**
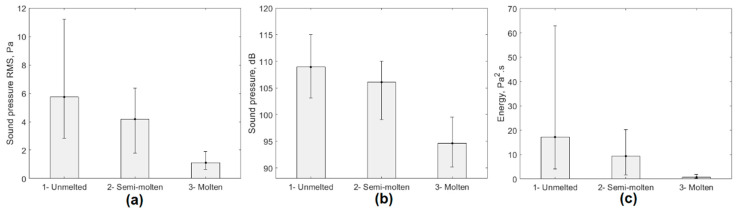
ANOVA of the behavior of the filtered acoustic signal. (**a**) RMS; (**b**) sound level; (**c**) signal energy.

**Figure 9 sensors-20-06840-f009:**
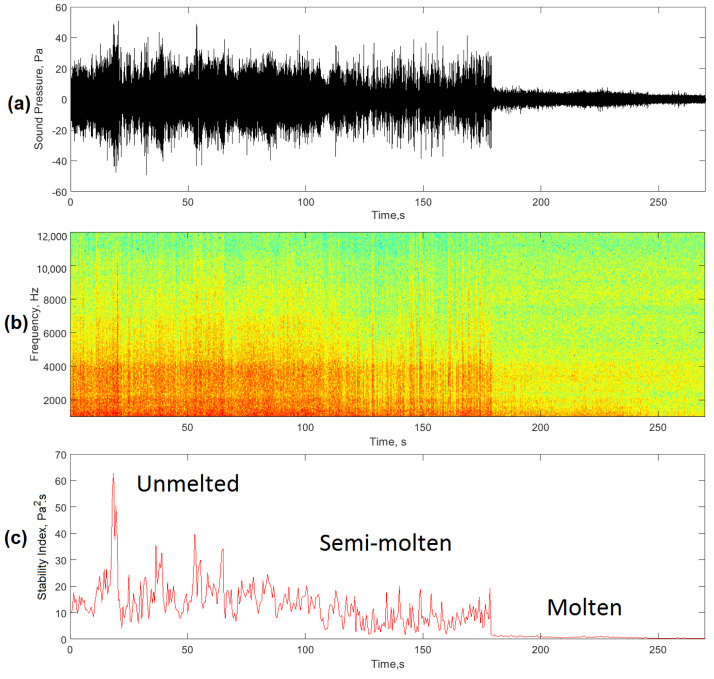
(**a**) Filtered acoustic signal (**b**) time-frequency diagram of the evolution of the filtered acoustic signal; (**c**) stability index values *AESI_EAF._*

**Figure 10 sensors-20-06840-f010:**
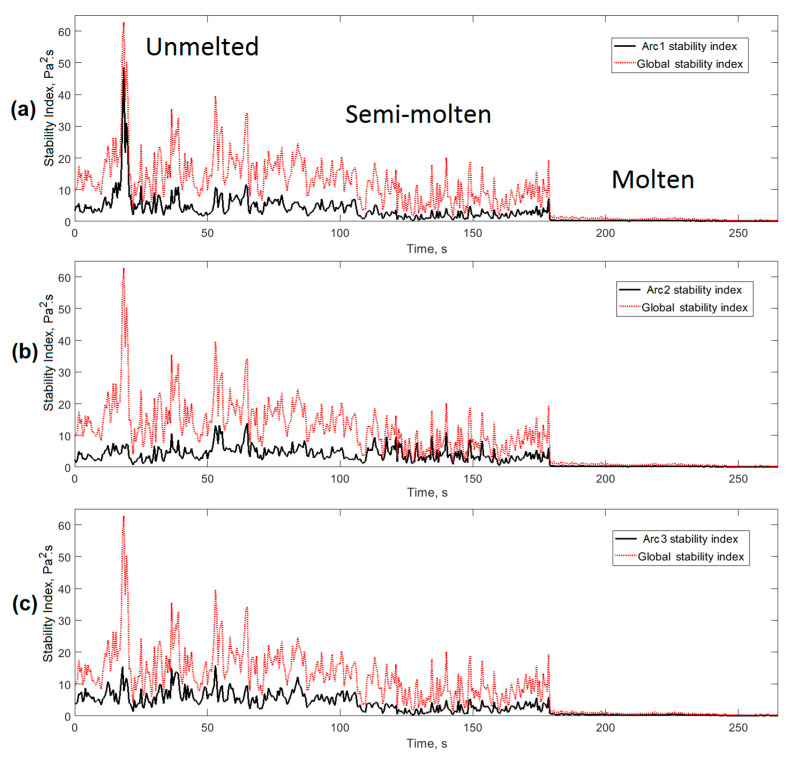
Influence of each phase on the overall stability of the EAF. (**a**) phase 1; (**b**) phase 2; (**c**) phase 3.

**Figure 11 sensors-20-06840-f011:**
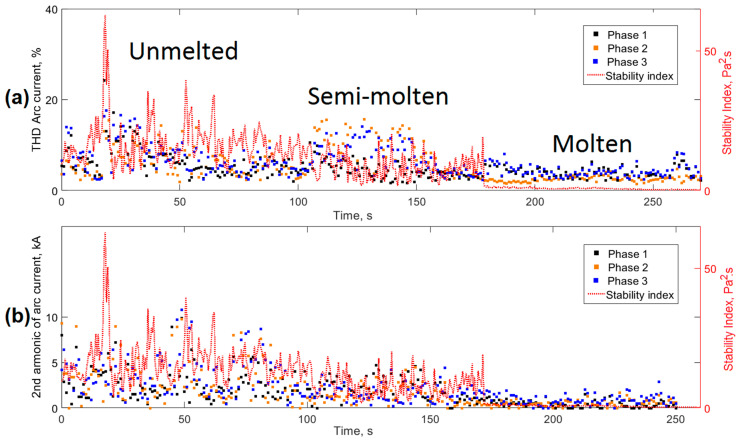
Relationship between the proposed stability index *AESI_EAF_* and the harmonics of the current signals. (**a**) Total harmonic distortion; (**b**) Second harmonic of the arc current.

**Figure 12 sensors-20-06840-f012:**
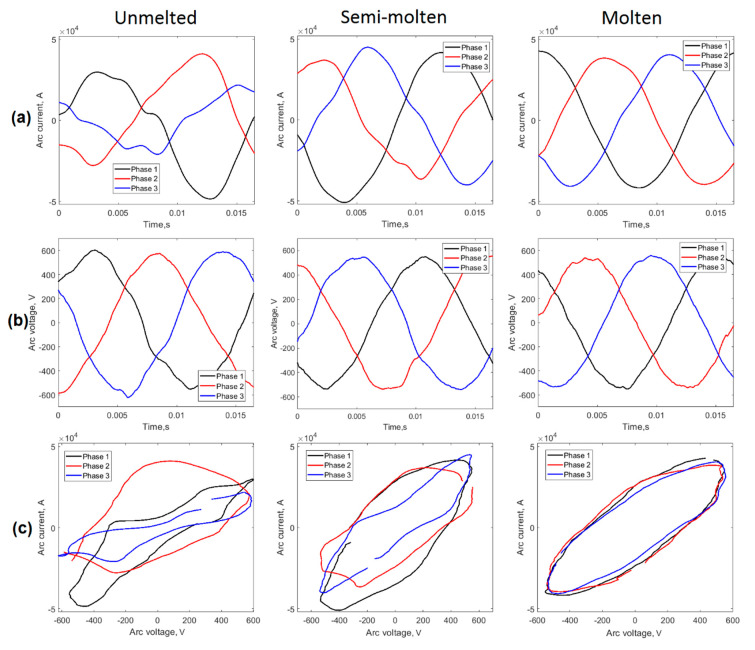
Behavior of the U and I signals in the three stages of the casting process in the EAF. (**a**) arc current; (**b**) arc voltage; (**c**) U-I characteristics.

**Figure 13 sensors-20-06840-f013:**
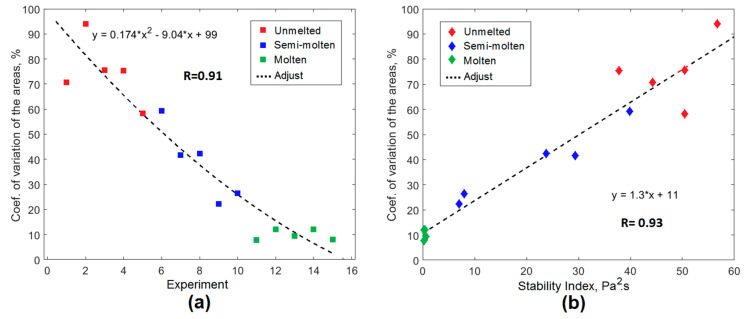
Correlation between U-I characteristics and new proposed *AESI_EAF_* index. (**a**) coefficient of variation of the areas of the U-I loops; (**b**) relationship between the coefficient of variation of the areas of the U-I loops and the *AESI_EAF_* stability index.

**Figure 14 sensors-20-06840-f014:**
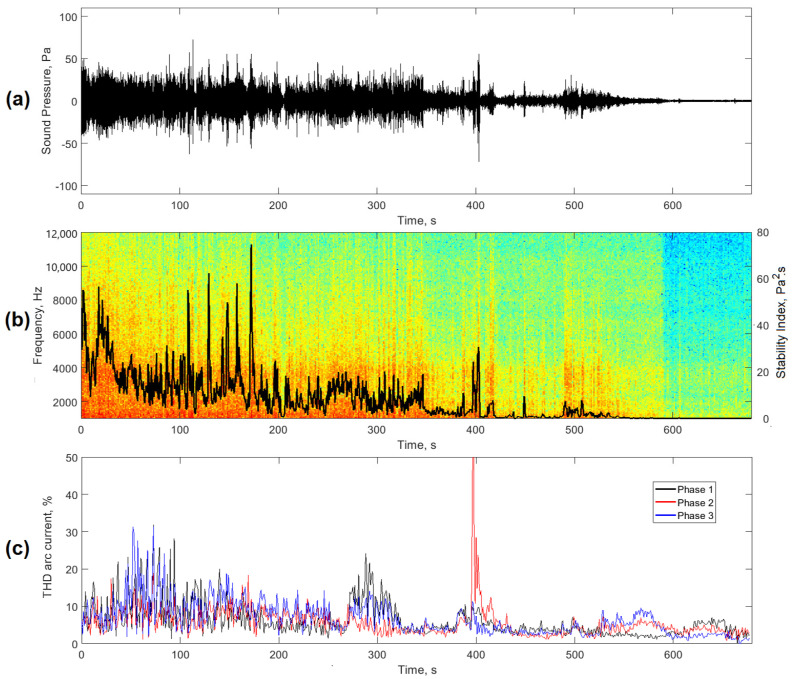
Application of the *AESI_EAF_* index to evaluate the stability of the process in a basket. (**a**) filtered acoustic signal; (**b**) spectrogram and *AESI_EAF_* index; (**c**) THD of arc currents.

**Table 1 sensors-20-06840-t001:** Main characteristics of the electric arc furnace (EAF).

Item	
Capacity:	60 ton
Furnace diameter	5.1 m
Electrodes diameter	0.5 m
Transformer power	40 MVA + 8%.
Power supply	AC three-phase 60 Hz
Electrode current per phase	30–45 kA
Transformer TAP	21
Raw materials	Scrap

**Table 2 sensors-20-06840-t002:** Acoustic signal descriptors.

Experiment	RMS (Pa)	Sound Level (dB)	Peak–RMS Ratio	Energy (Pa^2^·s)
Unmelted	24.14	121.94	4.51	319.10
Semi-molten	20.03	119.94	5.95	203.78
Molten	8.47	112.07	5.22	38.99

**Table 3 sensors-20-06840-t003:** *AESI_EAF_* Index Behavior.

Experiment	Average Values (Pa^2^·s)	Standard Deviation
**Unmelted**	17.24	8.45
**Semi-molten**	9.33	4.61
**Molten**	0.67	0.35

**Table 4 sensors-20-06840-t004:** Characterization of U-I loops.

Experiment	Area U_1_I_1_	Area U_2_I_2_	Area U_3_I_3_	CV (%)
**Unmelted**	2.08	5.42	1.49	70.75
	5.30	1.30	1.03	93.98
	1.02	2.51	0.54	75.62
	0.94	4.03	1.54	75.38
	2.35	4.38	1.31	58.22
**Semi-molten**	1.34	1.99	4.16	59.31
	1.12	1.84	2.69	41.63
	2.16	1.86	3.94	42.37
	1.73	2.73	2.23	22.28
	1.26	1.58	2.13	26.49
**Molten**	4.02	4.06	3.52	7.75
	3.29	4.11	3.46	12.02
	3.47	4.09	3.51	9.36
	3.11	3.26	2.58	12.06
	3.44	3.96	3.46	8.10

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
