# Peer review of "New Arc Stability Index for Industrial AC Three-Phase Electric Arc Furnaces Based on Acoustic Signals"

_sensors, 2020, doi:10.3390/s20236840_

Round 1

Reviewer 1 Report

The paper "New Arc Stability Index for Industrial AC Three-Phase Electric Arc Furnaces based on Acoustic Signals " analyzes an intesteresting topic, and proposes an index, based on acustic signals, to assess the stability of the melting process in electric arc furnaces. The paper should be improved following these comments before being considered for publication:

  • Abstract is OK
  • Keywords are OK
  • Statements should be backed uo by references. Please, add a reference for the first Introduction paragraph
  • Add a reference for lines 53-55
  • Line 59: Please define AE abbreviation
  • Line 62 (and same for 64): Drouet and Nadeau
  • Line 76: The authors of this paper, or Cherednichenko et al., please clarify it.
  • The novelty of the paper is clearly explained in the introduction, with a proper state of the art review.
  • Add a reference for Danieli EAF
  • Figure 1 caption should be centered
  • Lines 132 to 139: Add references for the equipment and software
  • The network analyzer is not mentioned in lines 132-139. A reference should also be added. Power analyzer brand is only shown in the abstract?
  • Line 195: "by connecting a three-phase AR6 power analyzer for electrical networks to the EAF power lines", as shown in Figure 1
  • Line: 197, define U-I, although this abbreviation is commonly known
  • Line 255: power supply frequency = 60 Hz? I guess it is because measurements are performed in a Cuban factory
  • Eq.4 should be AESIEAF
  • Eq.4: fcf or fcs?. Same for line 354
  • Is the AESi index dimensionless?
  • Eq.5: should be AESIEAF
  • Figure12c: Arc voltage, s? Should be Arc voltage,V?
  • Limitations and future research lines could be added

Author Response

Dear reviewer, below you will find an answer to all comments you have rightly made about the paper called New Arc Stability Index for Industrial AC Three-Phase Electric Arc Furnaces based on Acoustic Signals. The current version of the paper has been edited with the track changes activated and you can see every modification easily. 

The changes made to the document are detailed in the attached file (WORD).

Reviewer 2 Report

This method is clear from the experimental study.

The contribution of authors is well explained.

Some remarks:

Line 182 - Inform if the signal has been normalized.

Line 385 - Bibliographic reference missing.

Line 480 - Need to improve conclusion.

Author Response

(The authors gave the same response as above.)

Reviewer 3 Report

Regarding the paper entitled "New Arc Stability Index for Industrial AC Three-Phase Electric Arc Furnaces based on Acoustic Signals". The article is well written, and the contribution is sound. Some better explanation about the experiment set-up is required; for example, how the method is affected by changing the data acquisition system's sample time. Is any similar method reported in the literature that proposes a stability index?. If this is the case, can the authors compare their results?. The main contribution must have a better description, perhaps write the main contribution as an algorithm. Also, it is not totally clear the contribution; it looks like the main contribution is given only from the experimental part and the application of Equation (1).  Finally, the conclusion section does not consider future works or discussion about the method limitation.

Author Response

(The authors gave the same response as above.)

Round 2

Reviewer 1 Report

The authors have properly replied to all the reviewers' comments. Therefore, my final recommendation is Accept in present form.